# Salt Tolerance in *Machilus faberi*: Elucidating Growth and Physiological Adaptations to Saline Environments

**DOI:** 10.3390/biology13020075

**Published:** 2024-01-26

**Authors:** Qiong Mo, Yang Liu, Haohui Wei, Liyuan Jiang, En Wu, Ling Lin, Qihong Yang, Xiaoying Yu, Lihong Yan, Yanlin Li

**Affiliations:** 1College of Horticulture, Hunan Agricultural University, Changsha 410128, China; momo25@stu.hunau.edu.cn (Q.M.); liuyang1203@stu.hunau.edu.cn (Y.L.); weihaohui@stu.hunau.edu.cn (H.W.); w158.369-12_84@stu.hunau.edu.cn (E.W.); yqh@stu.hunau.edu.cn (Q.Y.); yuxiaoying@hunau.edu.cn (X.Y.); 2Hunan Botanical Garden, Changsha 410128, China; jly6493@163.com; 3Hunan Mid-Subtropical Quality Plant Breeding and Utilization Engineering Technology Research Center, Changsha 410005, China; 4Engineering Research Center for Horticultural Crop Germplasm Creation and New Variety Breeding, Ministry of Education, Changsha 410128, China; 5School of Economics, Hunan Agricultural University, Changsha 410128, China; lljjxy@hunau.edu.cn; 6School of Biological Sciences, Nanyang Technological University, 60 Nanyang Drive, Singapore 637551, Singapore

**Keywords:** salt stress, *Machilus faberi Hemsl*, adaptability, plant growth response, physiological indicators, biochemical indicators

## Abstract

**Simple Summary:**

In this study, the physiological response mechanism of *Machilus faberi Hemsl* under salt stress was discussed. In different salt concentration environments, we observed the growth status of and internal changes in plants. For example, plants of this species could still grow normally at a concentration of 100–300 mmol^−1^/L. These findings provide important theoretical data for the promotion of this species in saline–alkaline areas. This study provides a practical basis for the promotion of this species and fills the research gap of this species under abiotic stress.

**Abstract:**

Adversity stress is the main environmental factor limiting plant growth and development, including salt and other stress factors. This study delves into the adaptability and salt tolerance mechanisms of *Machilus faberi Hemsl*, a species with potential for cultivation in salinized areas. We subjected the plants to various salt concentrations to observe their growth responses and to assess key physiological and biochemical indicators. The results revealed that under high salt concentrations (500 and 700 mmol^−1^/L), symptoms such as leaf yellowing, wilting, and eventual death were observed. Notably, plant height and shoot growth ceased on the 14th day of exposure. Chlorophyll content (a, b, total a + b, and the a/b ratio) initially increased but subsequently decreased under varying levels of salt stress. Similarly, the net photosynthetic rate, stomatal conductance, leaf water content, and root activity significantly declined under these conditions. Moreover, we observed an increase in malondialdehyde levels and relative conductivity, indicative of cellular damage and stress. The activity of superoxide dismutase and ascorbate peroxidase initially increased and then diminished with prolonged stress, whereas peroxidase activity consistently increased. Levels of proline and soluble protein exhibited an upward trend, contrasting with the fluctuating pattern of soluble sugars, which decreased initially but increased subsequently. In conclusion, *M. faberi* exhibits a degree of tolerance to salt stress, albeit with growth limitations when concentrations exceed 300 mmol^−1^/L. These results shed light on the plant’s mechanisms of responding to salt stress and provide a theoretical foundation for its cultivation and application in salt-affected regions.

## 1. Introduction

Plants are often affected by a variety of abiotic stresses during their growth, with soil salinization being a critical environmental challenge that limits plant productivity [1]. Currently, approximately 800 million hectares of land globally, about 25% of the Earth’s land area, is affected by salinization, and the number is rising annually, especially in arid and semi-arid climates [2,3]. Factors such as low rainfall, high evaporation, poor water management, and the misuse of large amounts of fertilizers contribute to the increasing concentration of soil salinity [4]. Excessive salt in the soil leads to increased osmotic pressure in soil solutions, reduced soil aeration, and decreased water permeability. These changes not only diminish soil fertility but also significantly impede plant growth and productivity, potentially resulting in biodiversity loss [1,5].

Efforts to mitigate the negative impacts of salt stress on plants have led to significant advancements in enhancing plant salt tolerance through traditional selection and breeding techniques [2]. However, despite these advances, the survival adaptation mechanisms of plants under salt stress remain insufficiently understood [6]. The detrimental effects of salinity on plants can vary based on climatic conditions, light intensity, plant species, and soil properties. In response to salt stress, plants undergo a range of physiological and biochemical changes, such as osmoregulation, CO_2_ assimilation, photosynthetic electron transport, chlorophyll content and fluorescence, reactive oxygen species (ROS) production, and antioxidant defense systems [7]. Correlations have been observed between various biochemical indicators and the salt tolerance of plants [8]. Notably, most plants struggle to thrive in high-salt environments; concentrations as low as 100–200 mmol^−1^/L can hinder or halt their growth and development, potentially leading to plant death, sometimes within a short duration. However, a small number of plants survive in environments with a salt concentration of 300~500 mmol^−1^/L and show significant resilience [9,10]. Given the high salinity tolerance of these plants, there has been a growing interest in cultivating salt-tolerant species in salt-affected soils. This approach offers an alternative strategy for economic desalination and the restoration of severely degraded saline–alkaline lands [11]. This process, known as phytoremediation, leverages the inherent capabilities of these plants to rehabilitate salt-impacted environments effectively.

*Machilus faberi*, a member of the Lauraceae family’s *Machilus* genus, is a precious native tree species with wide application and unique ornamental value. Studying the physiological response mechanism of *M. faberi* is conducive to the popularization and application of this species. Therefore, *M. faberi* was selected for research. Existing studies have predominantly focused on the analysis of volatile oil composition in leaves [12], anti-fungi in vitro, transplanting technology [13], and forest land selection [14,15]. Research on the physiological responses of *M. faberi* under abiotic stress is relatively lacking.

The aim of this study was to explore the physiological traits, growth state, and photosynthetic system of *M. faberi* under salt stress, aiming to gain a comprehensive understanding of its salt tolerance mechanisms. This study evaluated the physiological response mechanism of salt stress on *M. faberi*. The results provide data support for its cultivation and application in saline environments.

## 2. Materials and Methods

### 2.1. Plant Materials

In May 2022, one-year-old seedlings of *M. faberi* exhibiting consistent growth and promising potential were sourced mainly from the germplasm resource garden of the Hunan Botanical Garden. These were planted in flowerpots (25 cm × 30 cm), using a cultivation matrix composed of garden soil, peat, perlite, and cow dung in a ratio of 8:6:3:1. The experimental site was located in the greenhouse of the Flower Base of Hunan Agricultural University (113°08′ E, 28°18′ N). The annual average temperature is 17.2 °C, the annual average precipitation is 1361.6 mm, the annual average sunshine is about 1200–1600 h, the daytime temperature is 25°~30°, the nighttime temperature is 8°~15°, and the relative humidity is 85%~95%. Unified water and fertilizer management was ensured.

### 2.2. Salt Stress Treatment Using Sodium Chloride

In this study, sodium chloride of high purity (greater than 99%) was utilized as the agent for salt stress treatment. Based on previous studies and pre-experimental results, we established five distinct salt concentration gradients: 0, 100, 300, 500, and 700 mmol^−1^/L. The plants grew strong and well. Each treatment group was set up with 3 replicates and 5 plants per replicate, and each plant was planted in a flowerpot. Different concentrations of salt solution were prepared, and salt treatment was performed every 5 days during the experiment. In order to avoid the loss of salt and water as much as possible, a tray was set at the bottom of the basin, and the exuded salt was poured back into the basin in time. In addition, in order to prevent salt accumulation, 300 mL of half-strength Hoagland nutrient solution was used to irrigate every three days [16].

### 2.3. Morphological Measurements

The data collected before the stress treatment were T0. On the 7th, 14th, 21st, and 28th days after stress treatment, the data were collected again as T7, T14, T21, and T28. The measured morphological characteristics included plant height, the growth length of new shoots, and the number of new shoots.

### 2.4. Chlorophyll Determination

Mature leaves were collected from the top of the plant and washed thoroughly with deionized water. The samples were dried and sliced on transparent paper. Then, 0.2 g of fresh leaf samples were weighed and soaked in 10 mL 95% ethanol for 24 h of dark incubation. Absorbance at 470 nm, 649 nm, and 665 nm was determined using an ultraviolet spectrophotometer (UNICO, Shanghai, China) [17]. This process was repeated at T0, T7, T14, T21, and T28 to determine chlorophyll content.

### 2.5. Determination of Pn and Gs

Net photosynthetic rate (Pn) and stomatal conductance (Gs) were measured using a portable photosynthesis system (LI-COR Bioscience, Lincoln, NE, USA). In each treatment, five new mature leaves were selected for testing. In order to ensure full activation, the test leaves were exposed to bright light for 10 min before testing.

### 2.6. Determination of Water Content, Relative Conductivity, Malondialdehyde Content, and Root Activity of Plant Leaves

The stressed material was washed, and the fresh weight of its leaves was weighed; then, it was dried in a 60° oven to constant weight, and its dry weight was measured.

The relative conductivity was assessed following Chen Aikui’s methodology to evaluate cell membrane integrity [18]. Uniformly sized plant leaves (ensuring leaf integrity and avoiding stem nodes) were first rinsed with tap water and then thrice with distilled water. Surface water was removed using filter paper, and the leaves were cut into strips of appropriate length, avoiding the main veins. Three fresh samples (0.1 g each) were quickly weighed and placed in a 10 mL graduated test tube containing deionized water, sealed with a glass stopper, and left to soak at room temperature for 12 h. The conductivity of the extract (R_1_) was measured using a conductivity meter. The samples were then boiled for 30 min, cooled to room temperature, shaken well, and the conductivity (R_2_) measured again. The relative conductivity was calculated as R_1_/R_2_ × 100%.

Oxidative damage to the cell membrane was assessed by measuring malondialdehyde (MDA) content changes over the duration of salt treatment. A 0.1 g sample was mixed with 1% trichloroacetic acid solution and centrifuged at 10,000× *g* for 10 min. Following this, 500 microliters of the supernatant was combined with an equal volume of 20% trichloroacetic acid solution containing 0.5% thiobarbituric acid. The mixture was heated at 90 °C for 20 min, followed by an ice bath to halt the reaction, and then centrifuged at 10,000× *g* for 5 min. Absorbance at 532 nm and 600 nm was measured, using a 20% trichloroacetic acid solution containing 0.5% thiobarbituric acid as a control [19]. Root activity was quantified using the TTC (triphenyl tetrazolium chloride) method [20].

### 2.7. The Activities of SOD, POD, and APX Were Determined

The activities of superoxide dismutase (SOD), peroxidase (POD), and ascorbate peroxidase (APX) in plant leaves were quantitatively assessed using enzyme activity assay kits provided by Sinobestbio.

### 2.8. The Contents of Pro, SS, and SP Were Determined

The free proline (Pro) in the leaves of *M. faberi* was detected via the sulfosalicylic acid method [21]. This technique is widely recognized for its accuracy in Pro quantification. For the assessment of soluble sugar (SS) content, anthrone colorimetry was employed, providing reliable measurements of sugar concentrations in plant tissues [22]. Additionally, the soluble protein (SP) content in the leaves of *M. faberi* was quantified using the Coomassie Brilliant Blue staining method [23], a standard approach known for its sensitivity and precision in protein analysis. Original manuscript data can be found in Appendix A.

### 2.9. Statistical Analysis

Data processing was conducted using Excel 2021 and IBM SPSS Statistics 26 software, while Origin 2021 and R4.3.1 was utilized for chart creation. A one-way analysis of variance (ANOVA) was employed to analyze the data of various indicators under different salt stress treatments, providing a robust statistical framework for evaluating the effects of salinity on plant physiology.

## 3. Results

### 3.1. Observation of Leaf Phenotype and Change in Growth Index

The results indicated that *M. faberi* exhibited varying responses to different concentrations of salt stress (Figure 1). Under low salt stress, leaf yellowing was observed starting from day 21, persisting until day 28. At a moderate salt concentration of 300 mmol^−1^/L, yellowing of leaves commenced on the 14th day, extending to the 28th day, with yellowing initiating from the leaf tips. In conditions of high salt stress (500 and 700 mmol^−1^/L), visible changes started from the 21st day, with leaves beginning to discolor from the tips and progressively wilting. By the 28th day, the leaves had completely withered. Moreover, salt stress at varying concentrations significantly impacted the number of shoots, shoot length, and plant height (Table 1). While the number of shoots declined under low-concentration stress, the shoot length and plant height continued to increase steadily. Conversely, under high salt stress, there was no increase in shoot length and plant height after 14 days. Combined with the phenotype, this is due to the deepening of the degree of stress, which causes plant growth disorder, resulting in no further growth.

### 3.2. Changes in Chlorophyll Content in Leaves of M. faberi

In this study, the leaves of *M. faberi* exhibited unique characteristics, and the contents of chlorophyll a and b did not decrease as the salt stress intensified. Instead, they peaked at values of 0.20 mg/g and 0.051 mg/g (Table 2), respectively, on the 21st day of treatment. A marked decline in chlorophyll content was observed on day 28, with the rate of decrease correlating with the salt stress concentration.

The chlorophyll a/b ratio can be used as an indicator of plant stress resistance. In general, lower chlorophyll a/b values indicate stronger stress resistance. The data in Table 2 showed that under various salt stress conditions, the chlorophyll a/b ratio in the leaves of *M. faberi* showed a downward trend, which was related to the increase in stress time and salt concentration.

### 3.3. Effects of Salt Stress on Pn and Gs

After 28 days of salt stress, the Pn of *M. faberi* at salt concentrations of 100, 300, 500, and 700 mmol^−1^/L decreased by 201%, 83.7%, 79.15%, and 95.37%, respectively, compared to the control group. Similarly, Gs experienced a significant reduction, decreasing by 98.77%, 99.31%, 99.56%, and 99.68% at the respective salt concentrations (Figure 2 and Appendix A).

### 3.4. Root Activity, MDA Content, and Electrical Conductivity in Response to Salt Stress

As the salt stress intensified, the root activity of *M. faberi* exhibited varying trends. Under salt stress of 100 and 300 mmol^−1^/L, the decrease in relative water content (RWC) was relatively modest. However, at higher concentrations of 500 and 700 mmol^−1^/L, RWC sharply declined, by 83.82% and 88.24%, respectively. Correspondingly, root activity showed a significant decrease, with reductions of 359%, 336%, 325%, and 276% observed as the salt concentration increased. Additionally, with the prolongation of salt stress, both malondialdehyde (MDA) content and relative electrical conductivity in *M. faberi* exhibited an upward trend (Figure 3 and Appendix A).

### 3.5. Enzymatic Responses in M. faberi Leaves under Salt Stress

The activity of SOD in *M. faberi* leaves exhibited a complex response to varying salt stress concentrations. Under low concentrations (100 and 300 mmol^−1^/L), SOD activity gradually increased. However, it decreased after 14 days of continuous exposure to high concentrations of salt stress, only to increase again after 28 days (Figure 4B and Appendix A). POD plays a crucial role in catalyzing the redox reaction between hydrogen peroxide (H_2_O_2_) and other substrates, mitigating the excess H_2_O_2_ produced by SOD and the membrane peroxidation damage caused by reactive oxygen species during stress. The POD activity in the leaves showed a consistent increase over the 28-day period under various salt stress levels (Figure 4A and Figure S4). Concurrently, APX activity also displayed an increasing trend after 28 days of continuous treatment under low-concentration salt stress, though it remained lower than the baseline (T0) levels. Notably, under high-concentration salt stress (700 mmol^−1^/L), APX activity first decreased and then increased, reaching a peak value of 215.39U g^−1^FW in the T28 period (Figure 4C and Appendix A).

### 3.6. Accumulation of Soluble Proteins, Proline, and Soluble Sugars in M. faberi under Salt Stress

The levels of SP and Pro in the leaves of *M. faberi* exhibited an upward trend across different salt stress concentrations over time (Figure 5A,B and Appendix A). Notably, salt stress markedly enhanced the concentrations of proline and soluble protein. For instance, after 28 days of stress, the levels of proline and soluble protein under various salt concentrations were significantly higher than those in the control group, showing increases of 132%, 1407%, 1664%, 1652%, and 982% for proline and 652%, 1127%, and 1299% for soluble protein, respectively. Conversely, the content of SS in the leaves initially decreased during the T7 period (Figure 5C and Appendix A) and subsequently began to rise as the duration of stress extended. Under low-concentration salt treatments, the soluble sugar content decreased in the T21 period and then increased. However, in high-salt-stress concentrations, the soluble sugar content continued to decrease until 28 days.

### 3.7. Principal Component Analysis, Correlation, and Regression Insights in M. faberi under Salt Stress

It can be seen from Figure 6 that there is a close correlation between the morphological Characteristics, chlorophyll content, photosynthetic system, osmotic regulation system, and antioxidant system of *M. faberi* (Figure 6 and Appendix A). Notably, SOD exhibited a strong negative correlation with SS and APX. Furthermore, a pronounced negative relationship was observed between the Pn and Pro, indicating that a decrease in Pn leads to an increase in Pro content. Stomatal conductance also showed a significant negative correlation with SP and Pro, suggesting an increase in SP and Pro as stomatal conductance decreases. In addition, a positive correlation emerged between POD and the number of new shoots, implying that the quantity of new shoots rises in tandem with increasing POD content.

The principal component analysis of the five sample groups revealed significant segregation within the first principal component (accounting for 57.7% of total variance) and the second principal component (constituting 18.2% of total variance) (Figure 7 and Appendix A). This distinction indicates substantial differences in the leaf properties of *M. faberi* across the control (CK) and varying salt concentrations (100, 300, 500, and 700 mmol^−1^/L).

To examine these correlations in greater detail, linear regression analysis was conducted on each significant index under control, semi-lethal, and lethal concentrations, based on the correlation patterns illustrated in Figure 8. This analysis revealed that both plant height and shoot length were significantly correlated with the duration of stress, as evidenced (Figure 8A,B and Appendix A). However, the relationship between chlorophyll content and the number of days under stress was relatively weak (Figure 8D–G and Appendix A). Interestingly, the Pn and Gs exhibited strong and significant correlations at lethal and semi-lethal concentrations (Figure 8H,I and Appendix A). Furthermore, SOD, Pro and SP also demonstrated similar correlation trends (Figure 8J,L,M and Appendix A). Conversely, POD and SS displayed a weak and nonsignificant correlation (Figure 8K,N and Appendix A).

## 4. Discussion

When plants are subjected to salt stress, plant growth and development are impaired [24,25], and their phenotypes and growth indicators show different trends at salt concentrations [26]. In general, salt stress can restrict the growth of plants and lead to significant changes in phenotypic characteristics [27]. Plants showed slower growth rates, lighter leaf color, and slower shoot growth in the face of different concentrations of salt stress, which is consistent with previous studies [24]. According to Sarker and Oba [28], this phenomenon is caused by increased osmotic pressure from increased salinity, which impedes the absorption and transport of water, resulting in a water deficit in plants, thus limiting growth and development.

Studies have shown that when plants are subjected to salt stress, the chloroplast structure is damaged and the production of chlorophyll protein–lipid complex is reduced [29], resulting in a general decrease in chlorophyll content [27,30,31]. This study found that the chlorophyll content of leaves showed a trend of increasing first and then decreasing, which may be due to the response of plants to environmental changes in the early stage of stress, but with the extension of time, chlorophyll synthesis gradually decreased. This may be due to the decrease in the light energy absorption capacity of chloroplasts under high salt stress, which leads to the decrease in chlorophyll content [32]. At the same time, salt stress also limits the photosynthetic rate and stomatal conductivity of plants [33,34,35]. This study found that under salt stress of different concentrations, the results of Pn and Gs are consistent and show a downward trend with the extension of stress time, which is consistent with the research results of Soliman [36]. These results indicated that salt stress had adverse effects on the chlorophyll content and photosynthesis of plants [33].

When plants are under salt stress, the dynamic balance mode of reactive oxygen species production and removal will be broken, and the rapid accumulation of reactive oxygen species will cause membrane lipid peroxidation [37], causing plant death and injury. Plants can use antioxidant enzymes, such as SOD, POD, and APX, to clear excess oxygen species (ROS) [38] such as hydrogen peroxide and superoxide anions [39,40], so as to slow down the damage ROS cause to the lipid peroxidation of the cell membrane [41]. In this experiment, after applying salt stress at different concentrations, the SOD activity of the leaves of *M. faberi* continued to rise at 100 mmol^−1^/L concentration, increasing first and then decreasing at 300 and 500 mmol^−1^/L concentrations; it then then increased at a concentration of 700 mmol^−1^/L. The reason for this may be that the change in the environment and the different stress concentrations in the stress process led to a great difference in SOD activity, but with the extension of time, the SOD activity was always higher than that of the control group. The activity of APX decreased first and then increased, which may be due to intracellular oxidative stress caused by salt stress [42], which can lead to the consumption of ascorbic acid, so that the activity of APX is inhibited. With prolonged stress, plants may activate genes associated with antioxidant defense, leading to APX protein synthesis, thereby increasing APX activity to improve the plant’s ability to fight oxidative stress. The continuous increase in POD activity indicated that under stress conditions, these enzymes could reduce the damage caused by removing reactive oxygen species, which was consistent with previous research results [41,43]. SOD, POD, and APX activities rose under salt stress, which indicated that the plant could balance active oxygen in vivo by increasing the activity of antioxidant enzymes, so as to improve the plant’s salt tolerance [44].

SS, SPand Pro are important osmoregulatory substances in plants [45]. The results showed that SS, SP, and Pro showed an increasing trend under salt stress, which indicated that the cell osmotic sites were maintained by accumulating osmotic regulatory substances such as SS, SP, and Pro [46]. Thus, the osmotic pressure of cells was improved, and the ability to absorb and retain water was maintained. This is consistent with the findings of AI-Farsi [47] and Xiaoe Liu [48].

The diversity analysis of physiological and biochemical indexes of plants under salt stress reflects the physiological response of plants under salt stress, which plays an irreplaceable role in the application and promotion of plants [49]. We found there was a certain correlation among these indicators. There was a significant negative correlation between photosynthetic indexes and SP and Pro, indicating that the content of SP and Pro increased with the decrease in Pn and Gs. A significant positive correlation existed between chlorophyll content and photosynthetic index. The decrease in chlorophyll content likely led to the decrease in the photosynthetic capacity of plants. This may be due to the damage to the chlorophyll structure of the plant under stress conditions, and the destruction of the internal capsule membrane, which then leads to a decrease in the photosynthetic capacity of the leaves. In order to resist the damage caused by salt stress, the osmotic adjustment system continuously accumulates osmotic substances to maintain the normal operation of the plant. From the principal component analysis, we can clearly observe that there are significant differences between different concentrations of salt stress, and these were caused by the different degrees of damage suffered by plants. In order to analyze the correlation of plants in more detail, we also carried out linear regression analysis. It can be found that the correlation between net photosynthetic rate and stomatal conductance was strong and significant at lethal and semi-lethal concentrations. At the same time, SOD, Pro, and SP also showed a similar correlation. It indicated that with the deepening of stress, the contents of SOD, Pro, and SP in the leaves of *M. faberi* increased continuously to resist adversity, which was similar to the response of predecessors under high salt stress [49,50].

## 5. Conclusions

In conclusion, this study showed that salt stress significantly affected the phenotype, photosynthetic system, and antioxidant response of *M. faberi*. Among them, chlorophyll content, photosynthetic rate, and stomatal conductance showed a downward trend, while MDA, SOD, POD, SP, Pro, and other indicators showed an upward trend, indicating that the growth status and characteristics of this species were constantly changing to resist stress when necessary. Studies have shown that *M. faberi* can grow normally at a concentration of 100–300 mmol^−1^/L. This provides data support for the promotion of this species in saline–alkaline areas.

## Figures and Tables

**Figure 1 biology-13-00075-f001:**
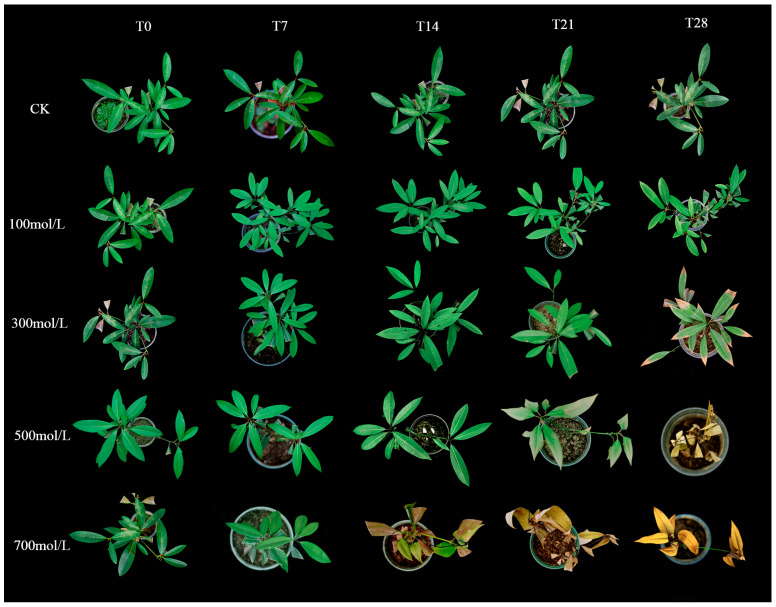
Leaf phenotype of *M. faberi* under different salt stress.

**Figure 2 biology-13-00075-f002:**
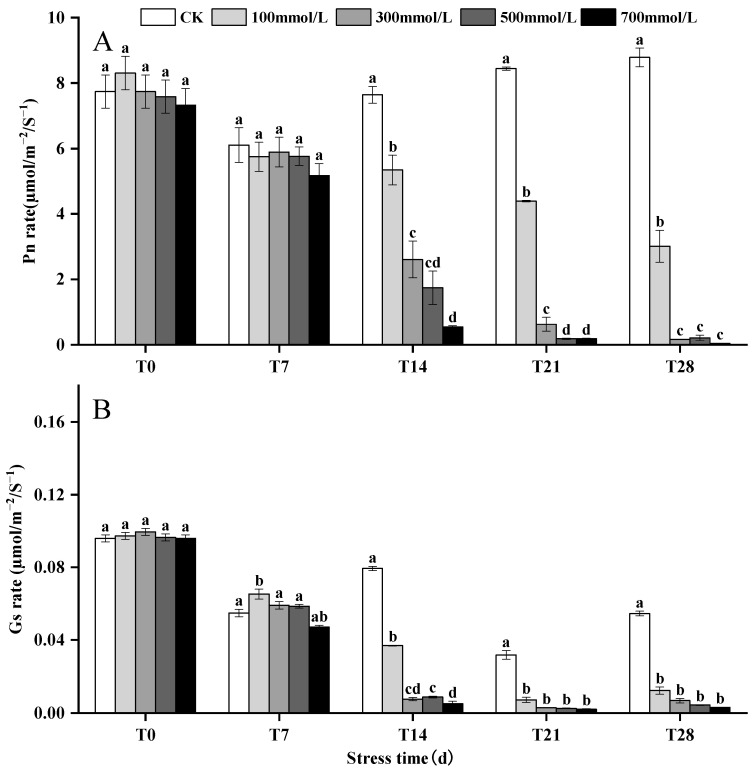
Effects of different concentrations of salt stress (0, 100, 300, 500, 700 mmol^−1^/L) on the Pn rate and Gs rate of leaves of *M. faberi* treated for 7, 14, 21 and 28 days. (**A**) Pn rate, (**B**) Gs rate. Different letters indicate differences between groups (*p* < 0.05).

**Figure 3 biology-13-00075-f003:**
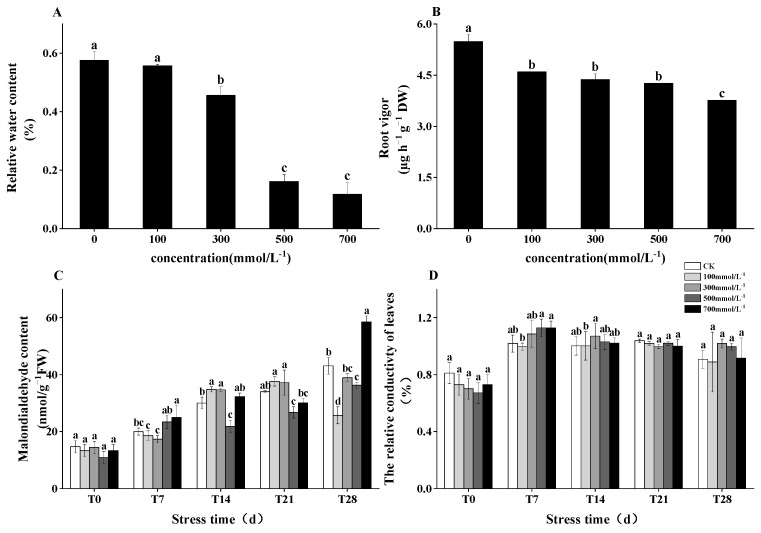
Variation trends in relative water content, root activity, malondialdehyde content and relative conductivity of leaves of *M. faberi* under different concentration gradients (0, 100, 300, 500, 700 mmol^−1^/L) and time (T0, T7, T14, T21, T28). The black columns in (**A**) and (**B**) represent period T28, (**C**) MDA content, (**D**) The relative conductivity of leaves. Different letters indicate significant differences between groups, *p* < 0.05 level.

**Figure 4 biology-13-00075-f004:**
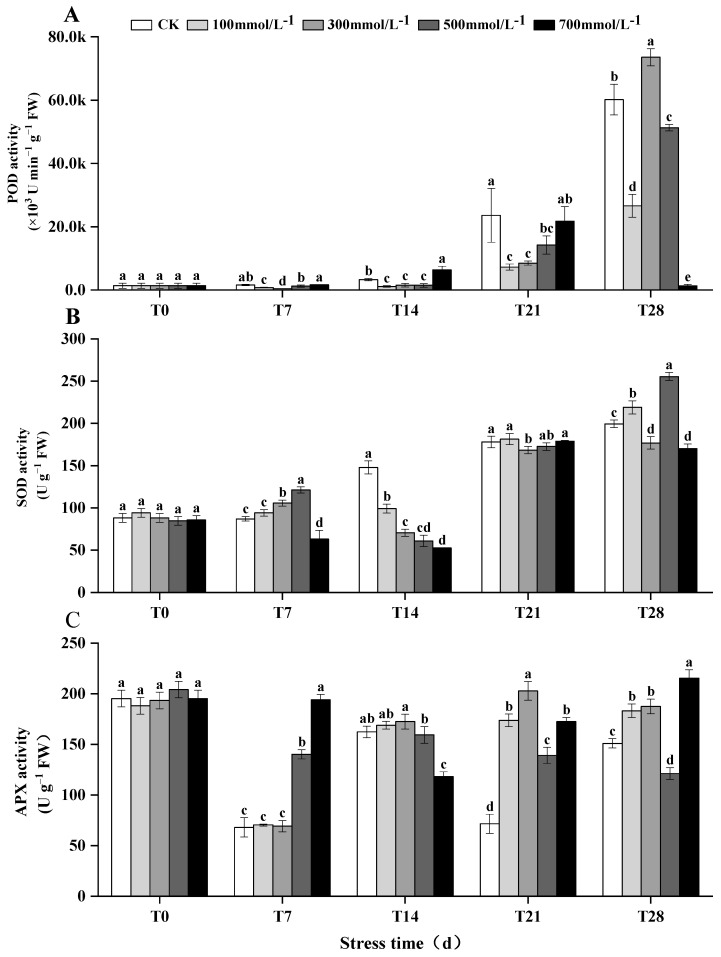
Effects of different concentrations of salt stress (0, 100, 300, 500, 700 mmol^−1^/L) on the activities of (**A**) POD, (**B**) SOD and (**C**) APX in leaves treated at 7, 14, 21 and 28 days. Different letters indicate differences between groups, *p* < 0.05.

**Figure 5 biology-13-00075-f005:**
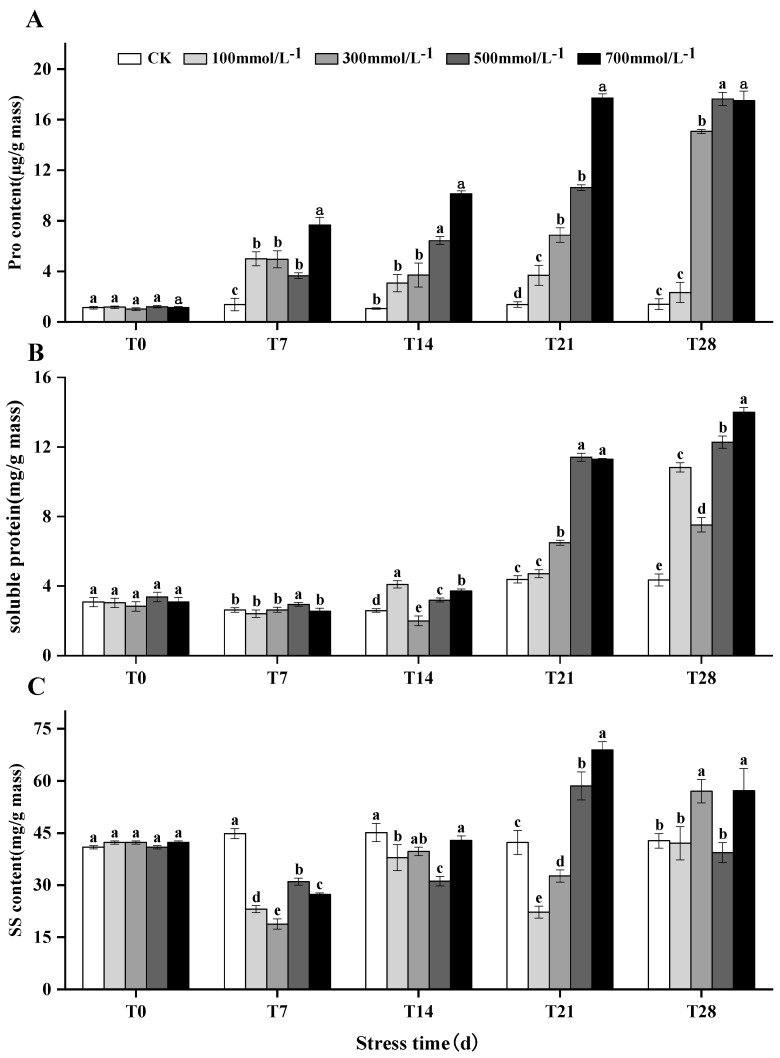
Effects of salt stress (0, 100, 300, 500, 700 mmol^−1^/L) on (**A**) proline, (**B**) soluble protein and (**C**) soluble sugar in leaves of *M. faberi* treated for 7, 14, 21 and 28 days. Different letters indicate differences between groups, *p* < 0.05.

**Figure 6 biology-13-00075-f006:**
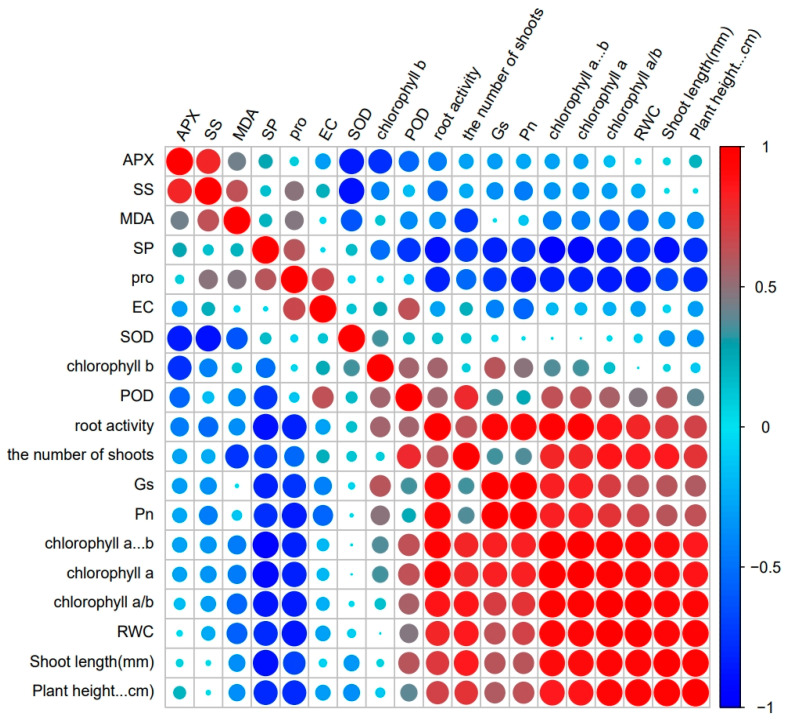
Correlation analysis of each index under salt stress. The positive correlation coefficient is represented by red, and the negative correlation index is represented by blue.

**Figure 7 biology-13-00075-f007:**
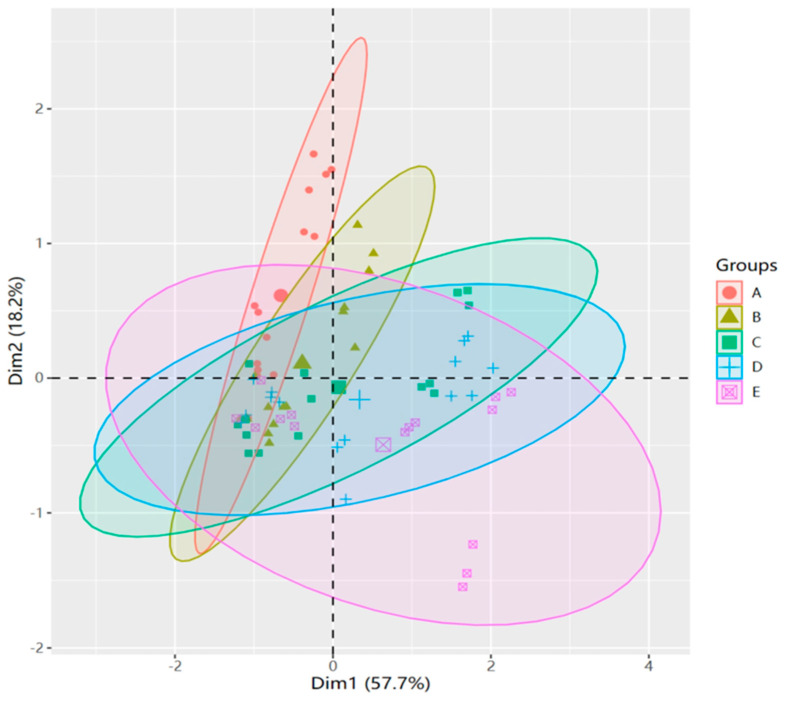
Principal component analysis between different salt concentrations. Group A is CK, group B is 100 mmol^−1^/L, group C is 300 mmol^−1^/L, group D is 500 mmol^−1^/L, and group E is 700 mmol^−1^/L.

**Figure 8 biology-13-00075-f008:**
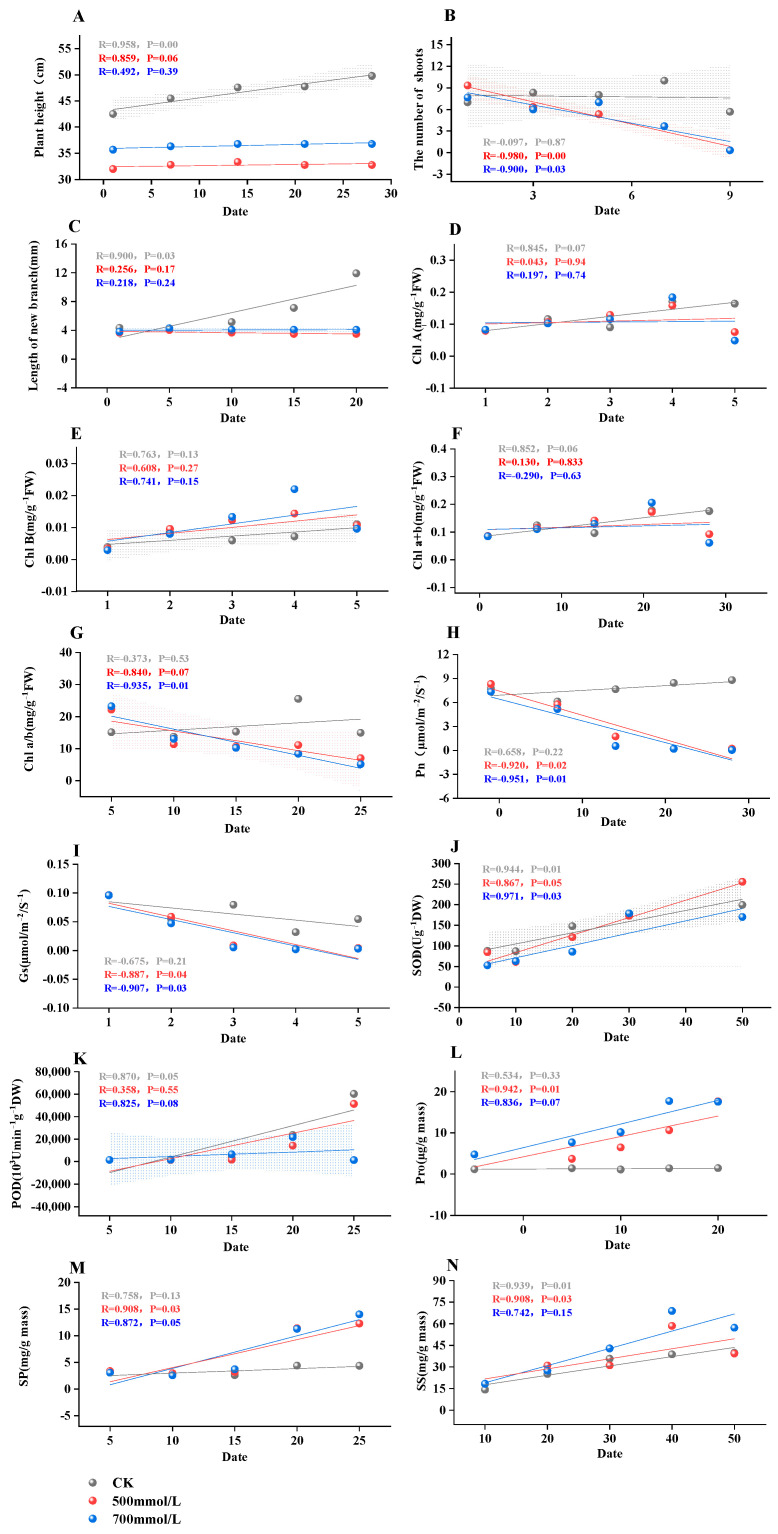
Studies the correlation analysis between CK, semi-lethal concentration (500 mmol^−1^/L) and lethal concentration (700 mmol^−1^/L) in the significant indicators of the correlation diagram. (**A**) Plant height; (**B**) The number of shoots; (**C**) Length of new branch; (**D**) Chl A; (**E**) Chl B; (**F**) Chl a+b; (**G**) Chl a/b; (**H**) Pn; (**I**) Gs; (**J**) SOD; (**K**) POD; (**L**) Pro; (**M**) SP; (**N**) SS. The Pearson correlation coefficient values are listed digitally below each correlation plot. The *p* value of each correlation pair was calculated by *t* test.

**Table 1 biology-13-00075-t001:** Morphological characteristics of *M. faberi* Hemsley on chinensis under salt stress.

Morphologic Characteristics	Days	T0	T7	T14	T21	T28
	Concentration (mmol^−1^/L)					
Number of new shoots (pcs)	CK	7.00 ± 1.73a	8.33 ± 1.89a	8.00 ± 1.46ab	10.00 ± 1.66a	5.67 ± 1.04a
	100	8.67 ± 1.53a	12.33 ± 1.11a	10.33 ± 1.21a	7.33 ± 1.13a	6.00 ± 1.00a
	300	13.00 ± 1.00a	11.33 ± 1.53a	10.33 ± 1.16a	8.00 ± 1.36	7.00 ± 1.73a
	500	9.33 ± 1.53a	6.33 ± 1.31a	5.33 ± 1.15ab	3.67 ± 1.52a	3.67 ± 1.52a
	700	7.67 ± 1.08a	6.00 ± 1.00a	1.00 ± 1.00b	0.33 ± 0.58a	0.33 ± 0.58a
Length of new branch (mm)	CK	4.32 ± 0.59a	4.23 ± 0.66a	5.13 ± 0.95a	7.10 ± 1.95a	16.94 ± 5.56a
	100	3.60 ± 0.17a	3.50 ± 0.51a	3.53 ± 1.25a	4.25 ± 0.50b	12.98 ± 5.11a
	300	3.58 ± 1.40a	7.39 ± 4.34a	3.82 ± 1.49a	3.81 ± 0.28b	16.39 ± 7.45b
	500	3.65 ± 0.94a	3.97 ± 0.19a	3.66 ± 0.92a	3.49 ± 1.18b	3.49 ± 1.18b
	700	3.79 ± 0.53a	4.26 ± 1.34a	4.05 ± 0.99a	4.05 ± 0.99b	4.05 ± 0.99a
Plant height (cm)	CK	42.51 ± 6.21a	45.47 ± 5.91a	47.56 ± 6.10a	47.77 ± 5.91a	49.77 ± 3.36a
	100	41.00 ± 8.77a	42.98 ± 8.501a	44.94 ± 8.71a	44.66 ± 7.64ab	48.13 ± 8.78ab
	300	39.97 ± 10.61a	43.00 ± 11.01a	44.11 ± 12.49a	43.37 ± 12.77ab	47.57 ± 10.37ab
	500	31.98 ± 3.94a	32.80 ± 4.10a	33.34 ± 4.20a	32.78 ± 3.61b	32.78 ± 3.61c
	700	35.67 ± 3.11a	36.31 ± 2.50a	36.79 ± 2.84a	36.79 ± 2.84ab	36.79 ± 2.84b

Note: Different letters indicate differences between groups (*p* < 0.05).

**Table 2 biology-13-00075-t002:** Chlorophyll content under different concentrations of salt stress.

Chl Species	Days	T0	T7	T14	T21	T28
	Concentration (mmol^−1^/L)					
	CK	0.081 ± 0.002a	0.115 ± 0.005a	0.090 ± 0.007c	0.171 ± 0.027b	0.164 ± 0.017a
	100	0.082 ± 0.002a	0.098 ± 0.007b	0.113 ± 0.002b	0.207 ± 0.009a	0.119 ± 0.001b
Chl a (mg/g)	300	0.102 ± 0.013a	0.096 ± 0.001b	0.126 ± 0.006a	0.129 ± 0.015c	0.117 ± 0.003b
	500	0.158 ± 0.022a	0.106 ± 0.010ab	0.129 ± 0.004a	0.158 ± 0.018c	0.075 ± 0.005c
	700	0.157 ± 0.005a	0.102 ± 0.005b	0.116 ± 0.004b	0.184 ± 0.004ab	0.049 ± 0.003d
	CK	0.004 ± 0.001a	0.009 ± 0.001a	0.006 ± 0.001c	0.007 ± 0.003d	0.011 ± 0.001a
	100	0.007 ± 0.003a	0.008 ± 0.002a	0.031 ± 0.002a	0.014 ± 0.004c	0.009 ± 0.001a
Chl b (mg/g)	300	0.006 ± 0.002a	0.008 ± 0.003a	0.004 ± 0.001c	0.051 ± 0.006a	0.010 ± 0.001a
	500	0.008 ± 0.003a	0.010 ± 0.002a	0.012 ± 0.002c	0.014 ± 0.003c	0.011 ± 0.001a
	700	0.008 ± 0.003a	0.008 ± 0.001a	0.013 ± 0.006c	0.022 ± 0.001b	0.010 ± 0.001a
	CK	0.085 ± 0.00a	0.124 ± 0.004a	0.096 ± 0.008c	0.178 ± 0.030b	0.175 ± 0.019a
	100	0.0892 ± 0.00a	0.107 ± 0.009b	0.144 ± 0.004a	0.222 ± 0.013a	0.129 ± 0.001b
Chl a + b (mg/g)	300	0.108 ± 0.001a	0.103 ± 0.004b	0.129 ± 0.006b	0.181 ± 0.021b	0.127 ± 0.005b
	500	0.166 ± 0.02a	0.116 ± 0.011ab	0.141 ± 0.006a	0.172 ± 0.020b	0.086 ± 0.006c
	700	0.165 ± 0.007a	0.110 ± 0.004b	0.129 ± 0.007b	0.206 ± 0.004ab	0.058 ± 0.004d
	CK	22.180 ± 0.616a	13.682 ± 0.931a	15.294 ± 0.195b	25.533 ± 0.368a	14.932 ± 0.468a
	100	13.655 ± 5.779a	12.244 ± 0.976a	3.654 ± 0.172c	15.184 ± 0.183b	13.009 ± 0.904b
Chl a/b (mg/g)	300	16.318 ± 2.242a	13.552 ± 0.686a	36.357 ± 0.750a	2.511 ± 0.015c	11.905 ± 0.015b
	500	23.080 ± 9.908a	11.348 ± 0.895a	10.649 ± 0.531b	11.081 ± 0.733b	7.017 ± 0.198c
	700	23.726 ± 10.786a	13.016 ± 0.374a	10.221 ± 0.800b	8.387 ± 0.449ab	5.105 ± 0.420d

Note: Different letters indicate differences between groups (*p* < 0.05).

## Data Availability

Data are contained within the article and the Appendix A.

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
