# Peer review of "Salt Tolerance in Machilus faberi: Elucidating Growth and Physiological Adaptations to Saline Environments"

_biology, 2024, doi:10.3390/biology13020075_

Round 1
Reviewer 1 Report
Comments and Suggestions for Authors
Dear authors, subheading 2.2 does not clearly state how the treatments with sodium chloride were carried out. Please describe in detail.
Line 105: It is necessary to add the reference for “Hoagland nutrient solution”.
Line 110: T7, T14, T21, and T28 stages should be deciphered at first mention.
Line 111: What are meant by phenotypic parameters in your study?
Line 112: Write the units of measurement for the parameters: plant height, the growth length of new shoots and in the Table 1 too.
Line 197: “Morphological indexes” are not a good term. It is better to write "morphological characteristics or parameters".
There is no need to refer to the figures and tables in the Results section in the Discussion section.
And yet, the understanding of the mechanisms of salt tolerance stated in the aim is not sufficiently proven in the manuscript. I would advise the authors to more precisely formulate the aim in accordance with the results.
Author Response
Dear Reviewer,
Thank you for reviewing our manuscript entitled “Salt Tolerance in Machilus faberi: Elucidating Growth and Physiological Adaptations to Saline Environments” (biology-2805463). The comments are of great help to improving the manuscript. We have studied the comments carefully and performed corresponding corrections in the revised manuscript. The point-by-point responses to the comments and suggestions are listed below.
We sincerely thank the editor and all reviewers for your valuable feedback that we have used to improve the quality of our manuscript. The reviewer's comments are listed in regular black font, and specific questions have been numbered. Our response is in blue font, and the manuscript revisions are highlighted in yellow.
Suggestions for Authors
Dear authors, subheading 2.2 does not clearly state how the treatments with sodium chloride were carried out. Please describe in detail.
Line 105: It is necessary to add the reference for “Hoagland nutrient solution”.
Line 110: T7, T14, T21, and T28 stages should be deciphered at first mention.
Line 111: What are meant by phenotypic parameters in your study?
Line 112: Write the units of measurement for the parameters: plant height, the growth length of new shoots and in the Table 1 too.
Line 197: “Morphological indexes” are not a good term. It is better to write "morphological characteristics or parameters".
There is no need to refer to the figures and tables in the Results section in the Discussion section.
And yet, the understanding of the mechanisms of salt tolerance stated in the aim is not sufficiently proven in the manuscript. I would advise the authors to more precisely formulate the aim in accordance with the results.
Responses to the reviewer (original comments by reviewer are in black)
Q1: Dear authors, subheading 2.2 does not clearly state how the treatments with sodium chloride were carried out. Please describe in detail.
Reply: We think this is an excellent suggestion. We have made the appropriate changes detailing how salt stress is carried out. As follows: “Different concentrations of salt solution were prepared, and salt treatment was performed every five days during the experiment. To avoid the loss of salt and water as much as possible, a tray was set at the bottom of the basin, and the exuded salt was poured back into the basin in time”. (The revised section is located on Page3 Line104-107)
Q2:Line 105: It is necessary to add the reference for “Hoagland nutrient solution”.
Reply: Thanks for your kind suggestions. We added the relevant references to the Hoagland nutrient solution. References are as follows:
KARIMI S, KARAMI H, VAHDATI K, et al. Antioxidative responses to short-term salinity stress induce drought tolerance in walnut [J]. Scientia Horticulturae, 2020, 267.
Q3:Line 110: T7, T14, T21, and T28 stages should be deciphered at first mention.
Reply: We sincerely thank you for your careful reading. We explain in detail how these periods came to be, modified as follows: “The data collected once before the stress treatment was T0. On the 7th, 14th, 21st and 28th days after stress treatment., the date were collected again as T7、T14、T21andT28.”(The revised section is located at Page3 Line112-113).
Q4:Line 111: What are meant by phenotypic parameters in your study?
Reply: Thank you for your careful reading. I am sorry that my expression is improper. I want to express the morphological characteristics of plants. I have made modifications and told them in more professional terms,“morphological characteristics”. (The revised section is located on Page 3, Line 114).
Q5:Line 112: Write the units of measurement for the parameters: plant height, the growth length of new shoots and in the Table 1 too.
Reply: We think this is an excellent suggestion. According to your suggestion, we have increased the unit of plant height, new shoot length and new shoot number to Table 1. (The revised section is located on Page5 Line191-192).
Table 1. Morphological characteristics of M. faberi Hemsley on chinensis under salt stress
morphologic character |
days |
T0 |
T7 |
T14 |
T21 |
T28 |
Concentration (mmol/L) |
||||||
Number of new shoots(pcs) |
CK |
7.00±1.73a |
8.33±1.89a |
8.00±1.46ab |
10.00±1.66a |
5.67±1.04a |
100 |
8.67±1.53a |
12.33±1.11a |
10.33±1.21a |
7.33±1.13a |
6.00±1.00a |
|
300 |
13.00±1.00a |
11.33±1.53a |
10.33±1.16a |
8.00±1.36 |
7.00±1.73a |
|
500 |
9.33±1.53a |
6.33±1.31a |
5.33±1.15ab |
3.67±1.52a |
3.67±1.52a |
|
700 |
7.67±1.08a |
6.00±1.00a |
1.00±1.00b |
0.33±0.58a |
0.33±0.58a |
|
length of new branch(mm) |
CK |
4.32±0.59a |
4.23±0.66a |
5.13±0.95a |
7.10±1.95a |
16.94±5.56a |
100 |
3.60±0.17a |
3.50±0.51a |
3.53±1.25a |
4.25±0.50b |
12.98±5.11a |
|
300 |
3.58±1.40a |
7.39±4.34a |
3.82±1.49a |
3.81±0.28b |
16.39±7.45b |
|
500 |
3.65±0.94a |
3.97±0.19a |
3.66±0.92a |
3.49±1.18b |
3.49±1.18b |
|
700 |
3.79±0.53a |
4.26±1.34a |
4.05±0.99a |
4.05±0.99b |
4.05±0.99a |
|
plant height(cm) |
CK |
42.51±6.21a |
45.47±5.91a |
47.56±6.10a |
47.77±5.91a |
49.77±3.36a |
100 |
41.00±8.77a |
42.98±8.501a |
44.94±8.71a |
44.66±7.64ab |
48.13±8.78ab |
|
300 |
39.97±10.61a |
43.00±11.01a |
44.11±12.49a |
43.37±12.77ab |
47.57±10.37ab |
|
500 |
31.98±3.94a |
32.80±4.10a |
33.34±4.20a |
32.78±3.61b |
32.78±3.61c |
|
700 |
35.67±3.11a |
36.31±2.50a |
36.79±2.84a |
36.79±2.84ab |
36.79±2.84b |
Q6:Line 197: “Morphological indexes” are not a good term. It is better to write "morphological characteristics or parameters".
Reply: We sincerely appreciate the valuable comments. We adjust “Morphological indexes” to “morphological characteristics”. (The revised section is on Page 5, Lines 191-192).
There is no need to refer to the figures and tables in the Results section in the Discussion section.
Reply: Thanks for the above suggestion. Based on your suggestion, We modified the text in the discussion section.(Page 14-15,Line308-310、320-323、327-329、356-357、363-364、374、376-377)
Reviewer 2 Report
Comments and Suggestions for Authors
My comments
This manuscript entitled, “Salt Tolerance in Machilus faberi: Elucidating Growth and
Physiological Adaptations to Saline Environments, describe an investigation to explore the key physiological and biochemical processes of tree species Machilus faberi under different salt concentrations. Results showed that the net photosynthetic rate, stomatal conductance, leaf water content, and root activity significantly declined, but malondialdehyde levels and relative conductivity increased under salt stress. The activity of superoxide dismutase and ascorbate peroxidase initially increased and then diminished with prolonged stress, whereas peroxidase activity consistently increased. The study suggested M. faberi possessed a degree of tolerance to salt stress. The results from this study provided a scientific reference for cultivation and application of M. faberi in salt-affected habitats.
The topic of this study is interesting and important in the fields of restoration and environmental science. The content of the current manuscript meet the scope of “Biology” for publication. But it seems to me that the manuscript is not prepared very well. No enough information was provided about the experimental design, samplings and measurements in this study. Many sentences were written too long to be understood. I suggest the manuscript be rejected in this version, but could be resubmitted again for peer-reviewing after rewriting and modification. Here, I would like to provide my comments in detail for consideration when the manuscript is modifies.
Line 18 (L18): this sentence, “one of the most detrimental”, needs to re-write.
L19, please provide the whole Latin name of “Machilus faberi”.
L25-30, all abbreviations should be deleted in the abstract. They are only showed up once in the abstract.
L44, “this Fig” ? It needs to re-write.
L63-64, “In contrast, surviving in environments with 300-500 mmol/L salt concentrations”. It is not a compete sentence.
L71-74, this sentence is too long to be understood. Please rewrite it.
L71-74, the authors provided limited information about the Machilus faberi, but more information and data are required to show why the author select this tree species for this study.
L76-77, what does this sentence mean? “While M. faberi has a relatively wide distribution in China, research on this species remains”.
L81-82, there ate two aims. What is the purpose (aim) of this study?
L81-87, I would like to see the objectives and hypotheses of this study.
L89, where was this study conducted? Please briefly provide information to describe the study area.
L90, what does it mean: “annual specimens”? Is it one-year-old seedlings?
L92, “2-gallon basins’? It should be a pot.
L93, it would be better if the basic of the ratio is 1, instead of 0.5. Thus, it should be 8:6:3:1.
L96, Please provide conditions set up in the greenhouse (such as light density, temperature, moisture etc in daytime and nighttime) during the study period.
L101-102, the authors should provide information about why they select these 5 salt concentrations in this study. Does the study area has acid-rains containing these salt concentrations? If so, please provide relative data and information.
L102, how do these 15 seedlings plant? Was one seedling planted in a pot?
L103-106, how often were the salt solutions used during the period of this study? How many solutions were used for each salt concentration each time?
L109-110, what do T0, T7, T14, T21, and T28 mean? Authors should provide the descriptions about these signs.
L112115, the sentence should deleted from the Materials and Methods section.
L118, “0.2 g 118 of fresh leaf sample”. How did the authors get the leaf sample? Thus, the authors need to provide a description in detail how to get leaf samples during the study.
L123-124, it would be better if “days 0, 7, 123 14, 21, and 28 of the salt treatment” is consistent with “T0, T7, T14, T21, and T28”.
L126, how many leaves are used to measure net photosynthetic rate (Pn) and stomatal conductance (Gs) for each time? Authors need to provide the complete name for all the abbreviations when they were presented at first time. It is separated from the abbreviations in the Abstract.
L129-132, this sentence should be deleted from the Materials and Methods section.
L137-138, how did the author get the dry weight?
L162-164, this sentence should be deleted from the Materials and Methods section.
L200-203, the sentences should be deleted from the Results section.
L209-211, the sentences should be deleted from the Results section.
L223-225, the sentences should be deleted from the Results section.
L290-293, this sentence should be deleted from the Results section.
L298-300, this sentence should be deleted from the Results section.
L333-338, this sentence is too long to be understood. It should be divided into several short sentences.
L346-352, again, this sentence is too long to be understood.
Comments on the Quality of English LanguageThe major problem is many sentences are too long to be understood in this manuscript.
Author Response
Dear Reviewer,
Thank you for reviewing our manuscript, "Salt Tolerance in Machilus faberi: Elucidating Growth and Physiological Adaptations to Saline Environments” (biology-2805463). The comments are of great help to improving the manuscript. We have studied the comments carefully and performed corresponding corrections in the revised manuscript. The point-by-point responses to the comments and suggestions are listed below.
We sincerely thank the editor and all reviewers for your valuable feedback that we have used to improve the quality of our manuscript. The reviewer's comments are listed in regular black font, and specific questions have been numbered. Our response is in blue font, and the manuscript revisions are highlighted in green.
Suggestions for Authors
This manuscript entitled, “Salt Tolerance in Machilus faberi: Elucidating Growth and Physiological Adaptations to Saline Environments, describes an investigation to explore the critical physiological and biochemical processes of tree species Machilus faberi under different salt concentrations. Results showed that the net photosynthetic rate, stomatal conductance, leaf water content, and root activity significantly declined, but malondialdehyde levels and relative conductivity increased under salt stress. The activity of superoxide dismutase and ascorbate peroxidase initially increased and then diminished with prolonged stress, whereas peroxidase activity consistently increased. The study suggested M. faberi possessed a degree of tolerance to salt stress. The results from this study provided a scientific reference for cultivation and application of M. faberi in salt-affected habitats.
The topic of this study is interesting and important in the fields of restoration and environmental science. The content of the current manuscript meet the scope of “Biology” for publication. But it seems to me that the manuscript is not prepared very well. No enough information was provided about the experimental design, samplings and measurements in this study. Many sentences were written too long to be understood. I suggest the manuscript be rejected in this version, but could be resubmitted again for peer-reviewing after rewriting and modification. Here, I would like to provide my comments in detail for consideration when the manuscript is modifies.
Line 18 (L18): this sentence, “one of the most detrimental”, needs to re-write.
L19, please provide the whole Latin name of “Machilus faberi”.
L25-30, all abbreviations should be deleted in the abstract. They are only showed up once in the abstract.
L44, “this Fig” ? It needs to re-write.
L63-64, “In contrast, surviving in environments with 300-500 mmol/L salt concentrations”. It is not a compete sentence.
L71-74, this sentence is too long to be understood. Please rewrite it.
L71-74, the authors provided limited information about the Machilus faberi, but more information and data are required to show why the author select this tree species for this study.
L76-77, what does this sentence mean? “While M. faberi has a relatively wide distribution in China, research on this species remains”.
L81-82, there ate two aims. What is the purpose (aim) of this study?
L81-87, I would like to see the objectives and hypotheses of this study.
L89, where was this study conducted? Please briefly provide information to describe the study area.
L90, what does it mean: “annual specimens”? Is it one-year-old seedlings?
L92, “2-gallon basins’? It should be a pot.
L93, it would be better if the basic of the ratio is 1, instead of 0.5. Thus, it should be 8:6:3:1.
L96, Please provide conditions set up in the greenhouse (such as light density, temperature, moisture etc in daytime and nighttime) during the study period.
L101-102, the authors should provide information about why they select these 5 salt concentrations in this study. Does the study area has acid-rains containing these salt concentrations? If so, please provide relative data and information.
L102, how do these 15 seedlings plant? Was one seedling planted in a pot?
L103-106, how often were the salt solutions used during the period of this study? How many solutions were used for each salt concentration each time?
L109-110, what do T0, T7, T14, T21, and T28 mean? Authors should provide the descriptions about these signs.
L112115, the sentence should deleted from the Materials and Methods section.
L118, “0.2 g 118 of fresh leaf sample”. How did the authors get the leaf sample? Thus, the authors need to provide a description in detail how to get leaf samples during the study.
L123-124, it would be better if “days 0, 7, 123 14, 21, and 28 of the salt treatment” is consistent with “T0, T7, T14, T21, and T28”.
L126, how many leaves are used to measure net photosynthetic rate (Pn) and stomatal conductance (Gs) for each time? Authors need to provide the complete name for all the abbreviations when they were presented at first time. It is separated from the abbreviations in the Abstract.
L129-132, this sentence should be deleted from the Materials and Methods section.
L137-138, how did the author get the dry weight?
L162-164, this sentence should be deleted from the Materials and Methods section.
L200-203, the sentences should be deleted from the Results section.
L209-211, the sentences should be deleted from the Results section.
L223-225, the sentences should be deleted from the Results section.
L290-293, this sentence should be deleted from the Results section.
L298-300, this sentence should be deleted from the Results section.
L333-338, this sentence is too long to be understood. It should be divided into several short sentences.
L346-352, again, this sentence is too long to be understood.
Responses to the reviewer (original comments by reviewer are in black)
This manuscript entitled, “Salt Tolerance in Machilus faberi: Elucidating Growth and Physiological Adaptations to Saline Environments, describes an investigation to explore the critical physiological and biochemical processes of tree species Machilus faberi under different salt concentrations. Results showed that the net photosynthetic rate, stomatal conductance, leaf water content, and root activity significantly declined, but malondialdehyde levels and relative conductivity increased under salt stress. Superoxide dismutase and ascorbate peroxidase activity initially increased and then diminished with prolonged stress, whereas peroxidase activity consistently increased. The study suggested M. faberi possessed a degree of tolerance to salt stress. The results from this study provided a scientific reference for the cultivation and application of M. faberi in salt-affected habitats.
The topic of this study is exciting and vital in the fields of restoration and environmental science. The content of the current manuscript meets the scope of “Biology” for publication. However, it seems that the manuscript is not prepared very well. Not enough information was provided about this study's experimental design, samplings and measurements. Many sentences were written too long to be understood. I suggest the manuscript be rejected in this version, but it could be resubmitted again for peer-review after rewriting and modification. Here, I would like to provide my detailed comments for consideration when the manuscript is modified.
Reply: Thank you for your careful reading; we have discussed, modified and embellished some sentences and improved the experimental design, sampling and measurement information. I will seriously revise my manuscript to improve its quality. Thank you again for your advice.
Q2:Line 18 (L18): this sentence, “one of the most detrimental”, needs to re-write.
Reply: Thanks for the above suggestion. We rewrite this sentence: "Adversity stress is the main environmental factor limiting plant growth and development, including salt and other stress factors”. (Page 1 Line24-25)
Q3:L19, please provide the whole Latin name of “Machilus faberi”.
Reply: Thanks for your careful checks. I'm sorry for my negligence; we changed “Machilus faberi” to “Machilus faberi Hemsl”.(Page1 Line26)
Q4:L25-30, all abbreviations should be deleted in the abstract. They only showed up once in the abstract.
Reply: This is an excellent suggestion. We have deleted the abbreviations in the abstract.(Page1 Line32-38)
Q5:L44, “this Fig” ? It needs to re-write.
Reply: Thanks for the above suggestion. we changed “this Fig” to “and the number is rising annually ” (Page1 Line50)
Q6:L63-64, “In contrast, surviving in environments with 300-500 mmol/L salt concentrations”. It is not a compete sentence.
Reply: Thanks for the above suggestion. We change “In contrast, surviving in environments with 300-500 mmol/L salt concentrations” to “However, a small number of plants survive in the 300 ~ 500 mmol / L salt concentration environment and show significant resilience”.(Page Line 69-70)
Q7:L71-74, this sentence is too long to be understood. Please rewrite it.
Reply: We sincerely appreciate the valuable comments. Per your suggestion,
we rewrote the sentence: “It is a precious native tree species with wide application and unique ornamental value. Studying the physiological response mechanism of M. faberi is conducive to the popularisation and application of this species”.(Page 2 Line 76-78)
Q8:L71-74, the authors provided limited information about the Machilus faberi, but more information and data are required to show why the author select this tree species for this study.
Reply: This is an excellent suggestion. Because this species has not been widely promoted and applied, the research content is less, so the information about this species is limited. The existing data is on page 2, Line 79-81.
Q9:L76-77, what does this sentence mean? “While M. faberi has a relatively wide distribution in China, research on this species remains”.
Reply: This is an excellent suggestion. We rewrite the sentence: “The research on the physiological responses of M. faberi under abiotic stress is relatively blank”.(Page2 Line81-82)
Q10:L81-82, there are two aims. What is the purpose (aim) of this study?
Reply: Thank you for your careful reading; we adjusted the content of this part. This study aims to provide data support for the popularisation and application of this species in saline-alkali land by studying the physiological response mechanism of this species under salt stress. (Page 2 Line85-87)
Q11:L81-87, I would like to see the objectives and hypotheses of this study.
Reply: This is an excellent suggestion. This study aims to provide data support for the popularisation and application of this species in saline-alkali land by studying the physiological response mechanism of this species under salt stress. (Page 2 Line85-87)
Q12:L89, where was this study conducted? Please briefly provide information to describe the study area.
Reply: This is an excellent suggestion. The research site is in the greenhouse of the flower base of the College of Horticulture, Hunan Agricultural University, Changsha, Hunan, China. We have added detailed information about the research site based on your suggestions. (Page 2 Line94-98)
Q13:L90, what does it mean: “annual specimens”? Is it one-year-old seedlings?
Reply: We sincerely thank you for careful reading.We changed “annual specimens” to “one-year-old seedlings”. (Page 2 Line90)
Q14:L92, “2-gallon basins’? It should be a pot.
Reply: This is an excellent suggestion. We have adjusted the description of the flowerpot. From “2-gallon basins” to “flowerpots(25cm×30cm)”. (Page 2 Line92)
Q15:L93, it would be better if the basic of the ratio is 1, instead of 0.5. Thus, it should be 8:6:3:1.
Reply: Thanks for your kind suggestions. We changed “4:3:1:0.5” to “8:6:3:1”. (Page 2 Line93)
Q16:L96, Please provide conditions set up in the greenhouse (such as light density, temperature, moisture, etc, in daytime and nighttime) during the study period.
Reply: This is an excellent suggestion. According to your suggestion, we describe the environmental conditions in the greenhouse in detail. Specifically, as follows: “The daytime temperature is 25 ° ~30 °, the nighttime temperature is 8 ° ~15 °, and the relative humidity is 85 % ~95 %. Unified water and fertiliser management”. (Page 2 Line93-98)
Q17:L101-102, the authors should provide information about why they select these 5 salt concentrations in this study. Does the study area has acid-rains containing these salt concentrations? If so, please provide relative data and information.
Reply: This is an excellent suggestion. We determined the concentration gradient range of salt stress through the pre-experimental results. The experimental treatment was carried out in a greenhouse, and the external rainfall would not affect the change of the concentration range in the plant soil.
Q18:L102, how do these 15 seedlings plant? Was one seedling planted in a pot?
Reply: This is an excellent suggestion. We added this part of the content: the plant is growing well, has strong growth, and is a single plant in flowerpots. (Page 3, Line 103)
Q19:L103-106, how often were the salt solutions used during the period of this study? How many solutions were used for each salt concentration each time?
Reply: This is an excellent suggestion. In view of your suggestions, we have detailed the frequency of use of each salt solution which specific as follows: “Different concentrations of salt solution were prepared, and salt treatment was performed every five days during the experiment. To avoid the loss of salt and water as much as possible, a tray was set at the bottom of the basin, and the exuded salt was poured back into the basin in time”. (Page 3 Line 104-107)
Q20:L109-110, what do T0, T7, T14, T21, and T28 mean? Authors should provide the descriptions about these signs.
Reply: We sincerely thank you for your careful reading. We describe T0, T7, T14, T21 and T28 in detail. Specific as follows: “The data collected once before the stress treatment was T0. On the 7th, 14th, 21st and 28th days after stress treatment., the date were collected again as T7、T14、T21andT28”. (Page 3 Line 112-113)
Q21:L112-115, the sentence should deleted from the Materials and Methods section.
Reply: This is an excellent suggestion. We have removed this sentence from the Material and Method.
Q22:L118, “0.2 g 118 of fresh leaf sample”. How did the authors get the leaf sample? Thus, the authors need to provide a description in detail how to get leaf samples during the study.
Reply: Thanks for your kind suggestions. According to your suggestions, we describe the process of obtaining samples in detail. The methods are as follows: “The fifth mature leaves were collected from the top of the plant and washed thoroughly with deionised water. The samples were dried and sliced on transparent paper. 0.2 g of fresh leaf samples were weighed and soaked in 10 ml 95 % ethanol for 24 hours of dark incubation”. (Page 3 Line 118-121)
Q23:L123-124, it would be better if “days 0, 7, 123 14, 21, and 28 of the salt treatment” is consistent with “T0, T7, T14, T21, and T28”.
Reply: This is an excellent suggestion. I'm sorry for my carelessness. We made some changes according to your suggestion. (Page 3 Line 122-123)
Q24:L126, how many leaves are used to measure net photosynthetic rate (Pn) and stomatal conductance (Gs) each time? Authors need to provide the complete name for all the abbreviations when they were presented at first time. It is separated from the abbreviations in the Abstract.
Reply: This is an excellent suggestion. Five new mature leaves were selected for each treatment, and the net photosynthetic rate and stomatal conductance were measured. And when the abbreviation first appears, we provide a complete name. (Page 3, Line 126-129)
Q25:L129-132, this sentence should be deleted from the Materials and Methods section.
Reply: Thank you for your careful reading. We have removed this sentence from the Material and Method.
Q26:L137-138, how did the author get the dry weight?
Reply: We sincerely appreciate the valuable comments. According to your suggestions, we describe how to obtain dry weight in detail. The details are as follows:“The stressed material was washed, and the fresh weight of its leaves was weighed, and then it was dried in a 60 ° oven to constant weight, and its dry weight was measured”. (Page 3 Line 133-134)
Q27:L162-164, this sentence should be deleted from the Materials and Methods section.
Reply: This is an excellent suggestion. We have removed this sentence from the Material and Method.
Q28:L200-203, the sentences should be deleted from the Results section.
Reply: This is an excellent suggestion. We have removed this sentence from the Results section.
Q29:L209-211, the sentences should be deleted from the Results section.
Reply: This is an excellent suggestion. We delete this sentence from the Result section.
Q30:L223-225, the sentences should be deleted from the Results section.
Reply: We sincerely appreciate the valuable comments. We delete this sentence from the Result section.
Q31:L290-293, this sentence should be deleted from the Results section.
Reply: This is an excellent suggestion. We delete this sentence from the Result section.
Q32:L298-300, this sentence should be deleted from the Results section.
Reply: Thank you for your careful reading. We have removed this sentence from the Results section.
Q33:L333-338, this sentence is too long to be understood. It should be divided into several short sentences.
Reply: This is an excellent suggestion. According to your suggestion, we modified this sentence to make it more streamlined: “Plants showed slower growth rate, lighter leaf colour and slower shoot growth in the face of different concentrations of salt stress, which is consistent with previous studies”. (Page 14 Line 311-313)
Q34:L346-352, again, this sentence is too long to be understood.
Reply: This is an excellent suggestion. According to your suggestion, we modify this sentence so that readers can better understand, as follows:“This may be due to the decrease of light energy absorption capacity of chloroplasts under high salt stress, which leads to the decrease of chlorophyll content”. (Page 14, Line 323-325)
Reviewer 3 Report
Comments and Suggestions for Authors
The salinity tolerance capacity of M faberi was evaluated by authors in the present study. The manuscript is drafted very nicely. The importance of the manuscript is bringing a native species in to salt tolerant conditions. The study framed in perfect design and executed. However, some important points are missing from the experiment.
Why only salinity tolerance was carried out? Whether plant grows only in saline areas?
How many genotypes of M faberi tested for salinity tolerance?
Authors need to clarify the basis levels of salinity considered for experimentation?
Analysis part of the experiments is very weak, it needs a inclusion of multivariate statistics to be included in the analysis.
The results of the experiment needs to be interpreted based on the utility of the species, general interpretations does not provide any implications in practical utility of results
The manuscript needs a round of English editing
Manuscript may be accepted after a round of major revision
Comments on the Quality of English LanguageMinor enlgish editing is required
Author Response
Dear Reviewer,
Thank you for reviewing our manuscript, "Salt Tolerance in Machilus faberi: Elucidating Growth and Physiological Adaptations to Saline Environments” (biology-2805463). The comments are of great help to improving the manuscript. We have studied the comments carefully and performed corresponding corrections in the revised manuscript. The point-by-point responses to the comments and suggestions are listed below.
We sincerely thank the editor and all reviewers for your valuable feedback that we have used to improve the quality of our manuscript. The reviewer's comments are listed in regular black font, and specific questions have been numbered. Our response is in blue font, and the manuscript revisions are highlighted in blue.
Suggestions for Authors
The salinity tolerance capacity of M. fabric was evaluated by the authors in the present study. The manuscript is drafted very nicely. The manuscript's importance is bringing a native species in to salt tolerant conditions. The study was framed in perfect design and executed. However, some important points are missing from the experiment.
Why only salinity tolerance was carried out? Whether plant grows only in saline areas?
How many genotypes of M. faberi tested for salinity tolerance?
Authors need to clarify the basis levels of salinity considered for experimentation?
Analysis part of the experiments is very weak, it needs a inclusion of multivariate statistics to be included in the analysis.
The results of the experiment needs to be interpreted based on the utility of the species, general interpretations does not provide any implications in practical utility of results
The manuscript needs a round of English editing
Manuscript may be accepted after a round of major revision
Responses to reviewer (original comments by reviewer are in black color)
Q1:The salinity tolerance capacity of M. faberi was evaluated by authors in the present study. The manuscript is drafted very nicely. The importance of the manuscript is bringing a native species in to salt tolerant conditions. The study framed in perfect design and executed. However, some important points are missing from the experiment.
Reply: We think this is an excellent suggestion. About the important point you made, we modified it in the conclusion part. (Page 15 Line386-393)
Q2:Why only salinity tolerance was carried out? Whether plant grows only in saline areas?
Reply: We sincerely appreciate the valuable comments. We conducted a salt tolerance test to understand whether the species can grow generally in saline-alkali land and provide some theoretical support for the subsequent promotion of the species. This species grows typically in broad-leaved forests of 800-1500, with a wide range of growth.
Q3:How many genotypes of M faberi tested for salinity tolerance?
Reply: Thank you very much for your question. We currently have only one species, which provides an excellent direction for our research. In the future, we may breed this species to enrich our germplasm resources.
Q4:Authors need to clarify the basis levels of salinity considered for experimentation?
Reply: We sincerely thank you for your careful reading. We determined the concentration gradient range of salt stress through the pre-experimental results. The experimental treatment was carried out in a greenhouse, and the external rainfall would not affect the change of the concentration range in the plant soil.
Q5:Analysis part of the experiments is very weak, it needs a inclusion of multivariate statistics to be included in the analysis.
Reply: Thanks for your kind suggestions. By reviewing the relevant literature and combining our data, we conducted the following statistical analysis :
- Correlation analysis showed a close correlation between morphological and physiological indexes, chlorophyll content, photosynthetic system, osmotic regulation system and antioxidant system.
- Principal component analysis: dimensionality reduction reduces multiple variables to a few independent variables, and the degree of dispersion of the samples is evaluated.
- Linear regression analysis: based on correlation analysis, the correlation between indicators is verified in more detail.
Reference:
Gao, G., M.A. Tester, and M.M. Julkowska, The Use of High-Throughput Phenotyping for Assessment of Heat Stress-Induced Changes in Arabidopsis. Plant Phenomics, 2020. 2020.
Biermann, R.T., et al., Discovering Tolerance—A Computational Approach to Assess Abiotic Stress Tolerance in Tomato Under Greenhouse Conditions. Frontiers in Sustainable Food Systems, 2022. 6.
Q6:The results of the experiment needs to be interpreted based on the utility of the species, general interpretations does not provide any implications in practical utility of results
Reply: We sincerely appreciate the valuable comments. Regarding what you said about interpreting the results based on the effects of species, we think it is more in line with the discussion section. (Page 14-15 Line317-319,326-328,343-345,359-360,367-375)
Q7:The manuscript needs a round of English editing
Reply: We sincerely thank you for your careful reading. We will improve and polish the manuscript.
Q8:Manuscript may be accepted after a round of major revision
Reply: We sincerely thank you for careful reading. We will overhaul the manuscript to improve its quality.
Round 2
Reviewer 3 Report
Comments and Suggestions for Authors
The authors made significant improvements in the manuscript. The manuscript may be accepted for publication.